# Have the concepts of 'anxiety' and 'depression' been normalized or pathologized? A corpus study of historical semantic change

Yu Xiao[1], Naomi Baes[1], Ekaterina Vylomova[2], Nick Haslam[1]*

**1** School of Psychological Sciences, The University of Melbourne, Melbourne, Australia, **2** School of Computing and Information Systems, The University of Melbourne, Melbourne, Australia

* nhaslam@unimelb.edu.au

## Abstract

Research on concept creep indicates that the meanings of some psychological concepts have broadened in recent decades. Some mental health-related concepts such as 'trauma', for example, have acquired more expansive meanings and come to refer to a wider range of events and experiences. 'Anxiety' and 'depression' may have undergone similar semantic inflation, driven by rising public attention and awareness. Critics have argued that everyday emotional experiences are increasingly pathologized, so that 'depression' and 'anxiety' have broadened to include sub-clinical experiences of sadness and worry. The possibility that these concepts have expanded to include less severe phenomena (vertical concept creep) was tested by examining changes in the emotional intensity of words in their vicinity (collocates) using two large historical text corpora, one academic and one general. The academic corpus contained >133 million words from psychology article abstracts published 1970–2018, and the general corpus (>500 million words) consisted of diverse text sources from the USA for the same period. We hypothesized that collocates of 'anxiety' and 'depression' would decline in average emotional severity over the study period. Contrary to prediction, the average severity of collocates for both words increased in both corpora, possibly due to growing clinical framing of the two concepts. The study findings therefore do not support a historical decline in the severity of 'anxiety' and 'depression' but do provide evidence for a rise in their pathologization.

## Introduction

Studies of concept creep [1] indicate that in recent decades many harm-related concepts in psychology have undergone semantic broadening. Concepts such as abuse, bullying, prejudice, and trauma have come to refer to an increasingly wide range of actions and experiences. In the 1970s, for example, bullying was defined as a specific kind of childhood peer aggression that was repeated, intentional, and perpetrated downward in a power hierarchy. Over time, these criteria have loosened. In addition to such prototypical cases, bullying now commonly refers to aggression that is unrepeated, unintentional, and directed upward or laterally rather than exclusively downward, and is invoked to refer to adult as well as childhood behavior. [1]

https://osf.io/u9mve/?view_only=
f877aa015c78454fb62d898e51b47158.

**Funding:** Australian Research Council Discovery
Projects awarded to Nick Haslam supported this
research (DP170104948 and DP210103984). The
funders had no role in study design, data collection
and analysis, decision to publish, or preparation of
the manuscript.

**Competing interests:** The authors have declared
that no competing interests exist.

documented this pattern of semantic inflation in several concepts and distinguished two main forms. Concepts creep horizontally when they stretch to refer to qualitatively different phenomena, such as when 'bullying' expands to include workplace misbehavior or to aggression carried out via social media rather than in person. They creep vertically when they expand to include less severe phenomena, such as when the threshold for identifying bullying is lowered by no longer requiring behavior to be repeated. In theory, concept meanings could shift vertically by excluding less severe phenomena, but by Haslam's definition this would represent concept deflation or contraction rather than creep. The two forms of concept creep may co-occur.

Research on concept creep, reviewed by Haslam et al. [2], points to several potential contributors and consequences of the phenomenon. Its drivers may include a declining objective prevalence of harm and a rising cultural sensitivity to harm, both resulting in less severe harms being recognized. Its consequences may be mixed. On the one hand, by recognizing a wider range of harms, concept creep may allow previously ignored forms of suffering to be legitimated and addressed, and previously tolerated forms of maltreatment to be problematized. Recognizing gambling problems as addictions, unconscious bias as prejudice, or online abuse as bullying signals that these undesirable phenomena should be taken seriously. On the other hand, concept creep may have some disadvantages. Broadening concepts of harm may trivialize those concepts [3], lead to disproportionate responses to relatively mild cases, and promote a sense of personal vulnerability and fragility [4, 5]. If the concept of trauma dilutes or normalizes to include everyday adversities, for example, the concept may become trivialized, victims of "small-t trauma" may seek unnecessary clinical interventions, and increasing numbers of people may identify as lastingly damaged by relatively innocuous events.

Concept creep occurs for a wide assortment of harm-related concepts, but it may be especially germane to concepts of mental ill-health. Broadened concepts of mental illness have been studied extensively under the headings of diagnostic inflation [6], pathologization [7], and psychiatrization [8, 9]. Critics of the Diagnostic and Statistical Manual for Mental Disorders (DSM), for example, have argued that successive editions have expanded the range of experiences and behaviors that are identified as mental disorders (horizontal concept creep), and that it has loosened the criteria for some disorders so that a lower severity threshold applies for their diagnosis (vertical concept creep). Although the evidence for wholesale diagnostic inflation is weak [10], there is strong evidence that certain diagnoses have expanded [e.g., ADHD: 11], with clear implications for over-diagnosis, over-medication, and resulting misallocation of clinical resources. Alongside any such expansion of concepts of mental ill-health within psychiatry, there are also concerns that laypeople's concepts may also be broadening, resulting in apparent epidemics of self-diagnosed conditions. In view of the timeliness of these concerns about the semantic broadening of mental health-related concepts, it is important to investigate them in academic and general discourse.

To date, research on the creep of mental health-related concepts has only addressed 'trauma' [12]. Studies have shown that this concept has risen steeply in use within academic psychology, and that this rise in usage has been accompanied by a broadening of the range of semantic contexts in which the term is employed (horizontal creep) [13, 14]. This research further indicates that the increase in the use of the term plays a causal role in its semantic inflation. A recent study by Baes, Vylomova, Zyphur, and Haslam [15] yielded similar findings for the vertical creep of 'trauma'. Using a novel method, Baes et al. showed that over a five-decade period beginning in the 1970s, 'trauma' came to be associated with less emotionally severe words within a large corpus of psychology article abstracts. Again, analysis indicated that the rising usage of 'trauma' contributed to this broadened meaning of the term, which

experimental studies suggest can lead to people judging 'trauma' to be less severe and less societally important [3].

In view of the paucity of research on the creep of mental health-related concepts beyond trauma, we conducted a study on possible semantic shifts in the concepts of anxiety and depression. It is particularly important to investigate shifts in these two concepts because they are central to some of the most common psychological problems in the general population and are often discussed together due to their symptomatic similarities [16] and comorbidity [17, 18]. Critics have argued that people experiencing normal worry, fear, sadness, or sorrow are increasingly diagnosed with anxiety and depressive disorders [19]. Horwitz and Wakefield [7] made the case that psychiatry had transformed everyday sadness into major depression, and Horwitz and Wakefield [20] extended their analysis to anxiety disorders, arguing that DSM commonly misdiagnoses adaptive anxieties as clinical conditions. By proposing that the diagnostic threshold for these disorders is too low, creating numerous false positives, this critique relates most strongly to the vertical form of concept creep. The controversy over the removal of the bereavement exemption for the diagnosis of major depression in DSM-5 [21] exemplifies this concern over the loosening of diagnostic criteria, as does DSM-5's [22] removal of the requirement that people receiving some anxiety disorder diagnoses must recognize that their anxiety is unreasonable.

Concerns over increasingly liberal diagnostic criteria for anxiety and depressive disorders relate to professional concepts of disorder, embodied in formal diagnostic systems. However, it is equally important to evaluate shifts in concepts of anxiety and depression in the wider culture. There is evidence that these concepts may have also undergone semantic inflation among laypeople. Bröer and Besseling [23], for example, argue that 'depression' can refer to ordinary sadness or low mood in colloquial language rather than exclusively to pathological conditions. If this is the case, broadened lay concepts of anxiety and depression might contribute to excessive self-diagnosis and inappropriate treatment [24, 25]. For example, people experiencing sub-clinical distress might strain mental health services, divert resources from people with more severe conditions [26], and experience side-effects from unnecessary medications.

Given evidence and concerns regarding the vertical concept creep of 'anxiety' and depression' in both professional and everyday language, we carried out a study of historical shifts in the emotional severity associated with each term in two large text corpora: professional-academic (psychology article abstracts) and general (a curated corpus of diverse USA-derived texts). Employing the methodology developed in Baes et al.'s [15] analysis of 'trauma', we examined whether words occurring in the vicinity of instances of 'anxiety' or 'depression' (i.e., collocates) tended to become less negative and intense in connotation–based on established norms for the affective meaning of words [27]–from 1970 to 2018. Consistent with the declining pattern observed for 'trauma' by Baes et al. [15] and our expectation that 'anxiety' and 'depression' have come to refer to increasingly mild and everyday experiences, we hypothesized that these concepts would appear in the context of less emotionally severe words over time (vertical concept creep). In addition to testing this primary hypothesis in each corpus, we planned to conduct post hoc analyses to clarify whether specific collocates of 'anxiety' and 'depression' might account for any observed trends.

## Method

### Materials

The psychology corpus comprised 871,340 abstracts from 875 psychology journals, collected from E-Research and PubMed databases, covering the period 1930 to 2019 [28]. The journal set was distributed broadly across the psychology discipline, including the following number

of journals from each of the following, non-mutually exclusive Scimago journal classifications: 216 (24.7%) developmental and educational psychology, 171 (19.5%) clinical psychology, 158 (18.1%) social psychology, 156 (17.8%) psychology (miscellaneous), 144 (16.5%) applied psychology, 99 (11.3%) experimental and cognitive psychology, and 45 (5.1%) neuropsychology and physiological psychology. Abstracts were used due to copyright restrictions and the fact that abstracts can effectively capture the main content of scientific articles [29]. The final corpus of psychology abstracts was limited to 1970 to 2018 data, due to the relatively small number of abstracts outside this bracket, yielding 133,082,240 words.

The general corpus was constructed by combining two existing corpora: the Corpus of Historical American English [CoHA; 30] and the Corpus of Contemporary American English [CoCA; 31]. CoHA contains 400 million words from the 1810s to the early 2000s, drawn from 115,000 texts evenly distributed across several kinds of everyday publication, including fiction, magazines, newspapers, and non-fiction books. CoCA contains 560 million words from 1990 to 2019 drawn from approximately 500,000 texts extracted from spoken language, TV shows, academic journals, fiction, magazines, newspapers, and blogs. To prevent potential overlap with the psychology corpus, we excluded CoCA texts that were sourced from academic journals, as well as excluding blogs due to their missing year data, before merging CoCA texts with CoHA to form a general corpus. This combined corpus has previously been demonstrated to be reliable [13]. We further extracted texts containing the phrase 'Great Depression' from both corpora in an effort to restrict usages of 'depression' to its psychiatric meaning. The phrase was rare (0.5% of all texts containing 'depression') in texts in the psychology corpus but common (14.2%) in the combined CoHA/CoCA corpus. Finally, only CoHA/CoCA texts between 1970 and 2018 were extracted to match the psychology corpus time period. In total, "anxiety" and "depression" appeared 47,324 times and 52,010 times, respectively, in the psychology corpus, and 7,959 times and 7,878 times in the CoHA/CoCA corpus (excluding texts with instances of "Great Depression"). Fig 1 presents the relative frequency of these centre terms by year in the two restricted corpora. The two centre terms appear much more frequently in the psychology corpus and become more prevalent in it over the study period.

## Warriner norms dataset

Affective meaning norms published by Warriner and colleagues [27] were used to evaluate the emotional severity of the contexts in which target words (i.e., 'centre terms') appeared. This dataset provides norms for valence, arousal, and dominance ratings of 13,915 English lemmas (i.e., the canonical or dictionary form of a set of word forms) provided by 1,827 United States residents (aged 16 to 87 years; 60% female). Participants rated how they felt while reading a word on a series of scales ranging from 1 (low) to 9 (high). For the valence rating ($n = 723$, $M = 5.1$, $SD = 1.7$), 1 corresponded to feeling extremely "annoyed", "bored", "despaired", "melancholic", "unhappy", or, "unsatisfied", and 9 corresponded to feeling extremely "contented", "happy", "hopeful", "pleased", or "satisfied". For the arousal rating ($n = 745$, $M = 4.2$, $SD = 2.3$), 1 represented feeling "calm", "dull", "relaxed", "sleepy", "sluggish", or "unaroused", while 9 indicated feeling "agitated", "aroused", "excited", "frenzied", "jittery", "stimulated", or "wide-awake". The dominance ratings were not used.

## Measures

**Contexts of the centre terms.**   The study evaluated semantic changes in the centre terms 'anxiety' and 'depression' by examining shifts in the emotional severity of collocated words occurring in their immediate context within the corpora. Following established practice [32], and consistent with Baes et al. [15], collocates were defined as individual words occurring

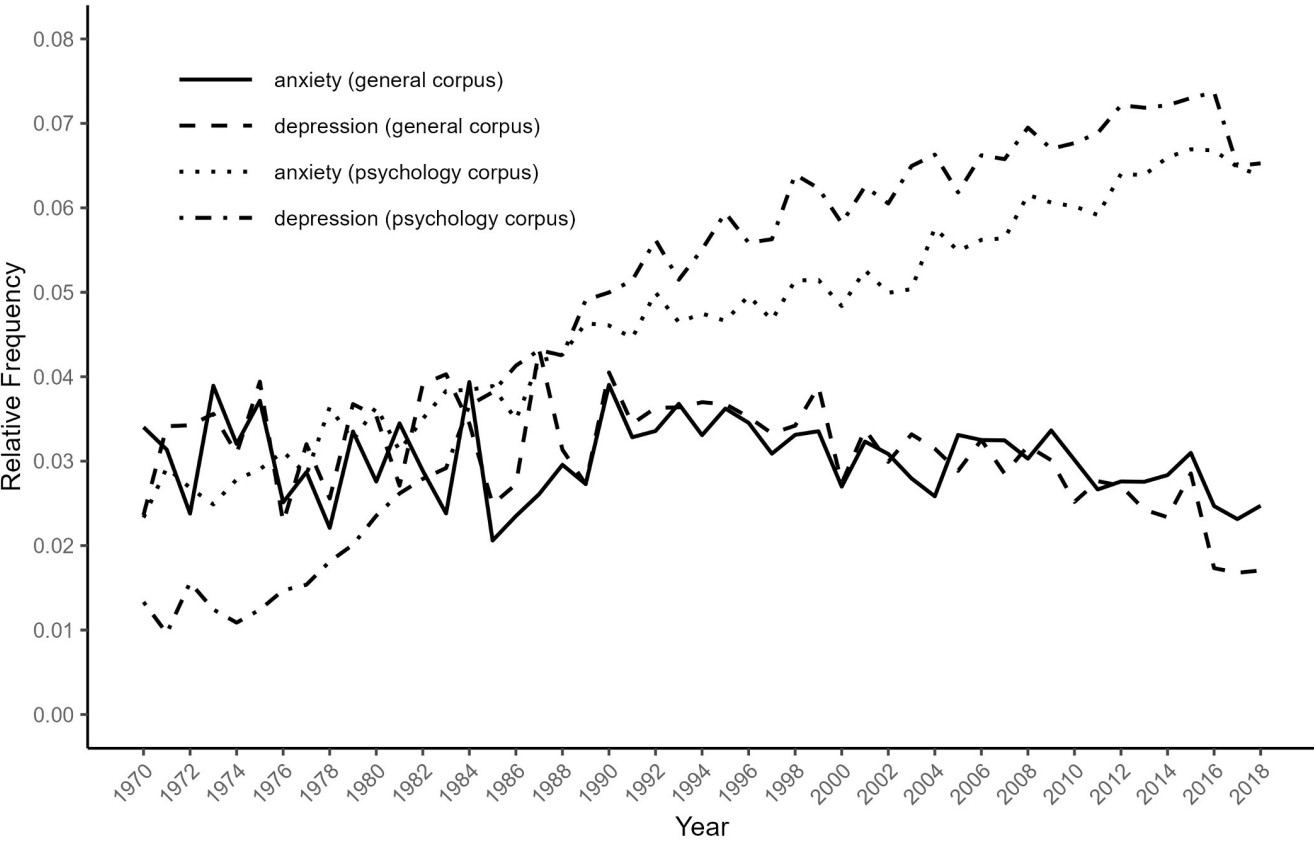

**Fig 1. Relative frequency of "anxiety" and "depression" in the corpora.**

within a ±5-word context window of each centre term. Before extracting the collocates, the text corpora were pre-processed by converting all words into their lemmas to reduce variations of word forms (e.g., "go" for "gone", "going" and "went"). Including repetitions of specific words, the collocate extraction procedure in the psychology corpus (1970–2018) resulted in 1,030,314 anxiety collocates, and 1,093,926 depression collocates; in the general (CoHA/CoCA) corpus (1970–2018), there were 107,611 anxiety collocates and 118,542 depression collocates.

**Severity index.** To compute an index of emotional severity, through which annual changes in the mean severity of 'anxiety' and 'depression' could be evaluated, we followed the procedure developed by Baes et al. [15]. For each corpus, collocates of the centre terms were matched with the Warriner norms dataset, disregarding collocates absent from those norms. Valence and arousal ratings were then summed to generate an index of emotional severity for each collocate, ranging from 2 to 18. Valence ratings were reverse scored (i.e., 1 = happy, 9 = unhappy) while arousal ratings were not (i.e., 1 = calm, 9 = aroused). The summed index assigns low scores for words judged to be emotionally positive and calm, and high scores for those judged to be unpleasant and intense. The severity index was then computed for each corpus by taking the weighted average collocate severity for each year (i.e., weighted by number of repetitions for collocates appearing more than once in the year). This index therefore represents the mean emotional intensity of words collocated with 'anxiety' and 'depression' in a particular year and would be expected to decline from 1970 to 2018 according to the study hypothesis.

**Analytic strategy.** Linear regression was performed to test the hypothesis that the severity index would decline. To investigate patterns of semantic change in greater detail, these analyses were followed up by identifying the most common collocates of 'anxiety' and 'depression' for each of the five decades, beginning with the 1970s, given the low frequency of most collocates in particular years. By examining changes in the relative frequencies of specific collocates across the decades, drivers of any trends in the severity index could be ascertained. All figures and statistical analyses were produced and processed using R in RStudio [33]. See the Open Science Framework repository for R scripts and associated files used in the current study: https://osf.io/u9mve/?view_only=f877aa015c78454fb62d898e51b47158.

## Results

### Psychology corpus

**Anxiety.** There were 826,083 anxiety collocates (including repetitions of specific collocates) that matched words in the Warriner norms, representing 80.2% of the anxiety collocates. Their severity scores ranged from 3.4 to 14.8 on the 2 to 18 scale ($M = 7.9$, $SD = 1.7$). A linear model with the severity index as the outcome variable and year as the predictor was statistically significant, $F(1, 47) = 132.10$, $p < .001$, accounting for 73% of the variance in the severity index. Contrary to hypothesis, the severity index increased over time (see Fig 2), indicating a significant rise in the severity of words occurring in the vicinity of 'anxiety', $t(47) = 11.49$, $p < .001$, $\beta = 0.86$, 95% CI [0.71, 1.01].

The top 10 most frequent anxiety collocates for each decade in the psychology corpus are displayed in Table 1. The words 'depression', 'disorder', and 'symptom' had the highest

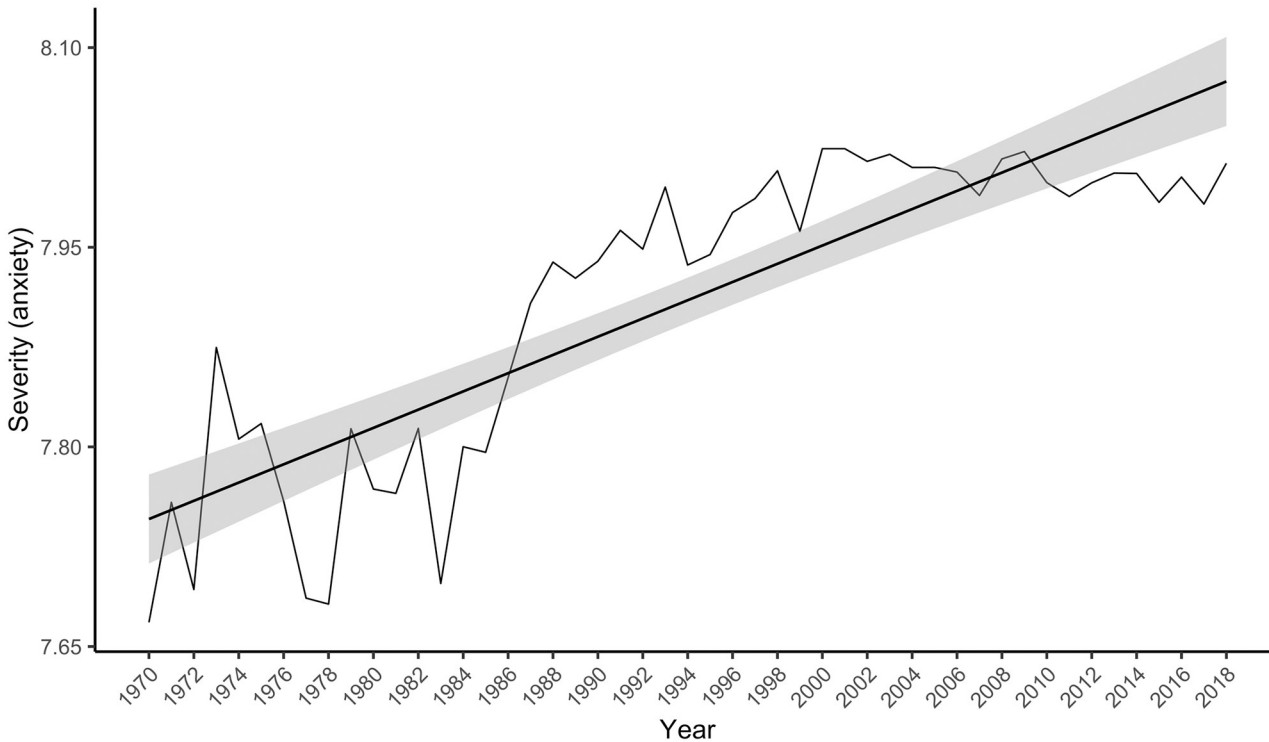

**Fig 2. The severity index for "anxiety" in the psychology abstracts corpus from 1970 to 2018.** The grey bars around the linear regression line indicate the standard error estimate.

**Table 1. Top 10 anxiety collocates in the psychology abstracts corpus by decade.**

| 1970s | 1980s | 1990s | 2000s | 2010s |
|---|---|---|---|---|
| test | depression | depression | depression | depression |
| state | trait | disorder | disorder | disorder |
| measure | state | high | symptom | symptom |
| trait | measure | trait | social | social |
| scale | high | state | high | high |
| level | test | symptom | child | study |
| high | self | measure | measure | child |
| group | level | self | level | associate |
| score | scale | patient | study | level |
| subject | report | social | trait | report |

severity ratings among these collocates, scoring 11.8, 10.1, and 9.8, respectively, all >1 SD above the mean severity scores of the anxiety collocates. These three words were among the most frequent collocates of 'anxiety' in every decade except the 1970s and therefore contributed substantially to the rise of the severity index from the 1970s to the 2010s. The relative frequencies of 'depression', 'disorder', and 'symptom' by decade are presented in Fig 3. All illustrate a statistically significant rising trend; for 'depression': β = 0.96, 95% CI [0.52, 1.42], $p$ = .006; for 'disorder': β = 0.93, 95% CI [0.28, 1.59], $p$ = .020; for 'symptom': β = 0.99, 95% CI [0.72, 1.25], $p$ = .001. This suggests that language surrounding 'anxiety' in the psychology abstracts became increasingly focused on anxiety's clinical and pathological aspects.

**Depression.** There were 854,235 Warriner-matched depression collocates in the psychology corpus, 78.1% of all the depression collocates, whose mean severity was 7.9 ($SD$ = 1.7). The linear model was statistically significant, $F(1, 47)$ = 9.47, $p$ = .003, accounting for 15% of the variance in the severity index. Again, contrary to the hypothesis the severity index rose (see Fig 4), rather than fell over time, $t(47)$ = 3.08, $p$ = .003, β = 0.41, 95% CI [0.14, 0.68].

The top 10 most frequent depression collocates for each decade are listed in Table 2. Collocates with the highest severity ratings were 'anxiety', 'disorder', and 'symptom', scoring 11.4, 10.1, and 9.8, respectively, all well above the mean for all collocates. 'Anxiety' and 'symptom' were among the most frequent collocates of 'depression' in every decade, and 'disorder' was frequent in three out of five decades. The relative frequencies of these common and influential collocates are presented by decade in Fig 5. 'Anxiety' and 'symptom' demonstrate a statistically significant increasing trend over time; for 'anxiety': β = 1.00, 95% CI [0.83, 1.15], $p$ < .001; for 'symptom': β = 0.94, 95% CI [0.34, 1.55], $p$ = .015; while 'disorder' is stable, β = 0.79, 95% CI [-0.35, 1.92], $p$ = .114.

## General corpus

**Anxiety.** There were 87,163 Warriner-matched anxiety collocates in the general (CoHA/CoCA) corpus, 73.5% of all the anxiety collocates, with an average severity score of 7.9 ($SD$ = 1.9). The linear model was statistically significant $F(1, 47)$ = 7.97, $p$ = .007, accounting for 13% of the variance in the severity index. Contrary to hypothesis, the severity index rose over time, $t(47)$ = 2.82, $p$ = .007, β = 0.38, 95% CI [0.11, 0.65], as Fig 6 shows.

The top 10 most frequent anxiety collocates for each decade are shown in Table 3. 'Fear', 'depression', and 'disorder' had the highest severity ratings among these collocates, scoring

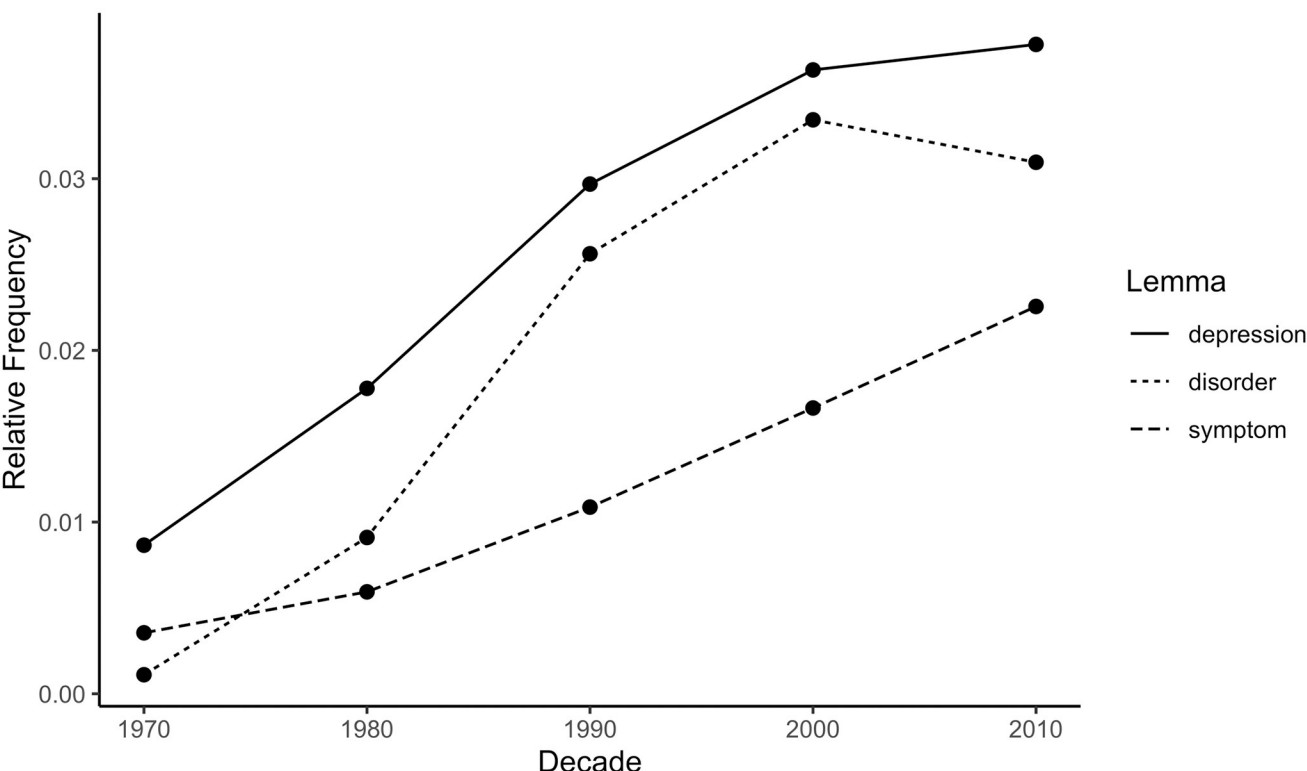

**Fig 3. Relative frequencies of selected anxiety collocates in the psychology abstracts corpus by decade.** Relative frequency is the summed repetitions of one lemma within a decade divided by the summed repetitions of all lemmas in the same decade. Larger relative frequency means higher frequency of the lemma in a particular decade.

12.2, 11.8, and 10.1, respectively. They were identified as among the top 10 most frequent collocates in at least two decades, therefore strongly influencing the overall severity index. Relative frequencies of 'fear', 'depression', and 'disorder' across the decades are plotted in Fig 7. The relative frequencies of 'depression' and 'disorder' both rose steeply with statistical significance; for 'depression': $\beta = 0.98$, 95% CI [0.58, 1.38], $p = .004$; for 'disorder': $\beta = 0.94$, 95% CI [0.31, 1.58], $p = .018$; whereas 'fear' remained relatively stable, $\beta = 0.61$, 95% CI [-0.84, 2.07], $p = .274$.

**Depression.** There were 95,476 Warriner-matched depression collocates in the general corpus, 88.7% of all the depression collocates, with an average severity score of 7.9 ($SD = 1.9$). The linear model was statistically significant, $F(1, 47) = 25.95$, $p < .001$, accounting for 34% of the variance in the severity index. Once again, mean severity increased over time (see Fig 8), contrary to hypothesis, $t(47) = 5.09$, $p < .001$, $\beta = 0.60$, 95% CI [0.36, 0.83].

The top 10 most frequent depression collocates by decade are presented in Table 4. There were several relatively frequent high severity collocates, notably 'war', 'suffer', 'problem', 'anxiety', and 'disorder'. Fig 9 shows that there was no consistent temporal trend for the first two terms; for 'war': $\beta = -0.72$, 95% CI [-2.00, 0.56], $p = .172$; for 'problem': $\beta = 0.68$, 95% CI [-0.66, 2.03], $p = .204$; whereas 'suffer' shows a statistically significant increase, $\beta = 0.94$, 95% CI [0.31, 1.57], $p = .018$. Fig 10 shows a statistically significant rise for the clinical terms 'anxiety' ($\beta = 0.94$, 95% CI [0.32, 1.56], $p = .017$) and 'disorder' ($\beta = 0.97$, 95% CI [0.53, 1.41], $p = .006$). As in earlier analyses, this pattern indicates an increase in the use of clinical or pathological language in the vicinity of 'depression' within the general corpus.

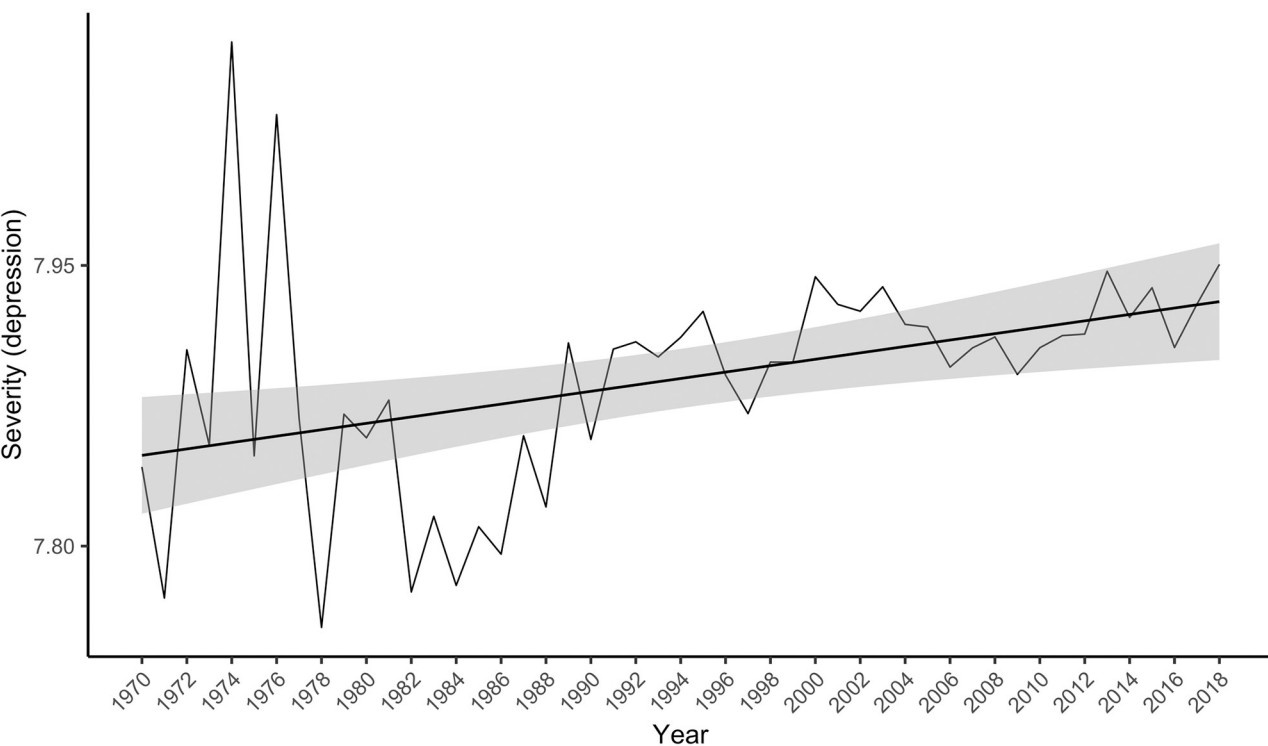

**Fig 4. The severity index for "depression" in the psychology abstracts corpus from 1970 to 2018.** The grey bars around the linear regression line indicate the standard error estimate.

## Discussion

The present study was the first to systematically examine long-term historical shifts in the meaning and use of 'depression' and 'anxiety'. From the theoretical standpoint of concept creep and using newly developed methods for evaluating changes in the emotional severity of word meanings, the study revealed consistent trends for the two concepts of interest across large text corpora representing academic-professional and general language use. These trends

**Table 2. Top 10 depression collocates in psychology abstracts corpus by decade.**

| 1970s | 1980s | 1990s | 2000s | 2010s |
|---|---|---|---|---|
| patient | anxiety | anxiety | anxiety | anxiety |
| scale | patient | patient | symptom | symptom |
| anxiety | scale | major | patient | study |
| measure | self | scale | scale | associate |
| group | measure | self | study | patient |
| symptom | inventory | symptom | major | scale |
| self | score | disorder | disorder | use |
| control | study | score | associate | disorder |
| study | symptom | measure | treatment | treatment |
| result | major | study | high | high |

Words were ranked by their relative frequency in each decade, from highest (top row) to lowest.

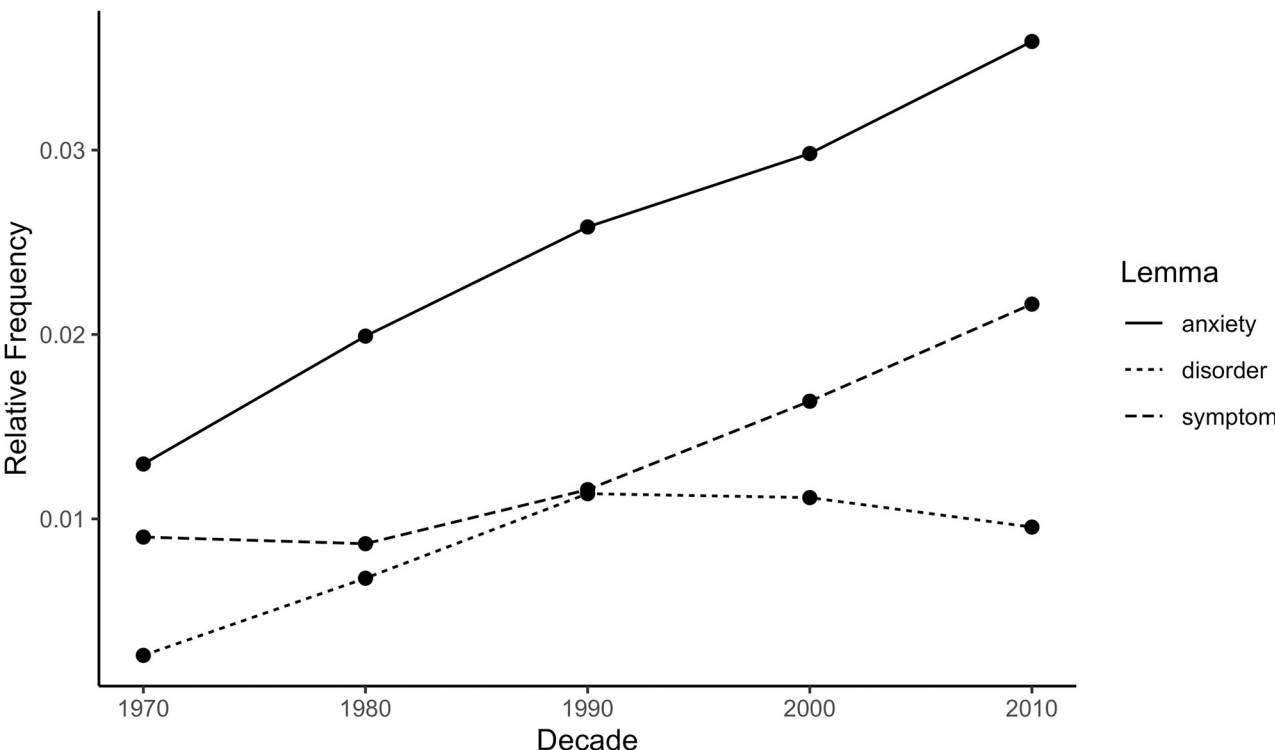

**Fig 5. Relative frequencies of selected depression collocates in the psychology abstracts corpus by decade.** Relative frequency is the summed repetitions of one lemma within a decade divided by the summed repetitions of all lemmas in the same decade. Larger relative frequency means higher frequency of the lemma in a particular decade.

represent linear increases in the emotional severity associated with 'anxiety' and 'depression' over the past half century.

These strong and consistent trends run contrary to the hypothesized direction of change. Based on well-established concerns that some mental health-related concepts have undergone semantic dilution in recent decades, reflected in diagnostic inflation within the mental health professions [6, 10] and colloquial use of clinical terms to reference everyday emotional states [23], we predicted 'anxiety' and 'depression' would decline in severity. If these concepts had crept vertically, their semantic context, represented by the words collocated with them, should have trended to become less severe over time, as Baes et al. [15] found for 'trauma' using an identical methodology. Instead, the average severity of collocates rose from the 1970s to the 2010s. The fact that this rise replicated across concepts and across very different corpora suggests that it is a robust effect rather than one confined to professional or general discourse.

Although our primary hypothesis was not supported, follow-up analyses offer some clues to what may have driven the observed rise in severity. Repeatedly, we found that many of the most common collocates of 'anxiety' and 'depression' were clinical terms and that these clinical collocates became more prevalent over the study period. In particular, the terms 'disorder' and 'symptom' tended to become more associated with 'anxiety' and 'depression' in more recent decades. In the psychology corpus, for example, neither term featured in the top 10 collocates in the 1970s and 1980s but 'disorder' became the second most common collocate from the 1990s through the 2010s and 'symptom' rose to third rank in the 2000s and 2010s. These patterns were almost equally striking in the general corpus. Although the collocates tended to be less clinical overall than in the psychology corpus and 'symptom' never featured in the top 10

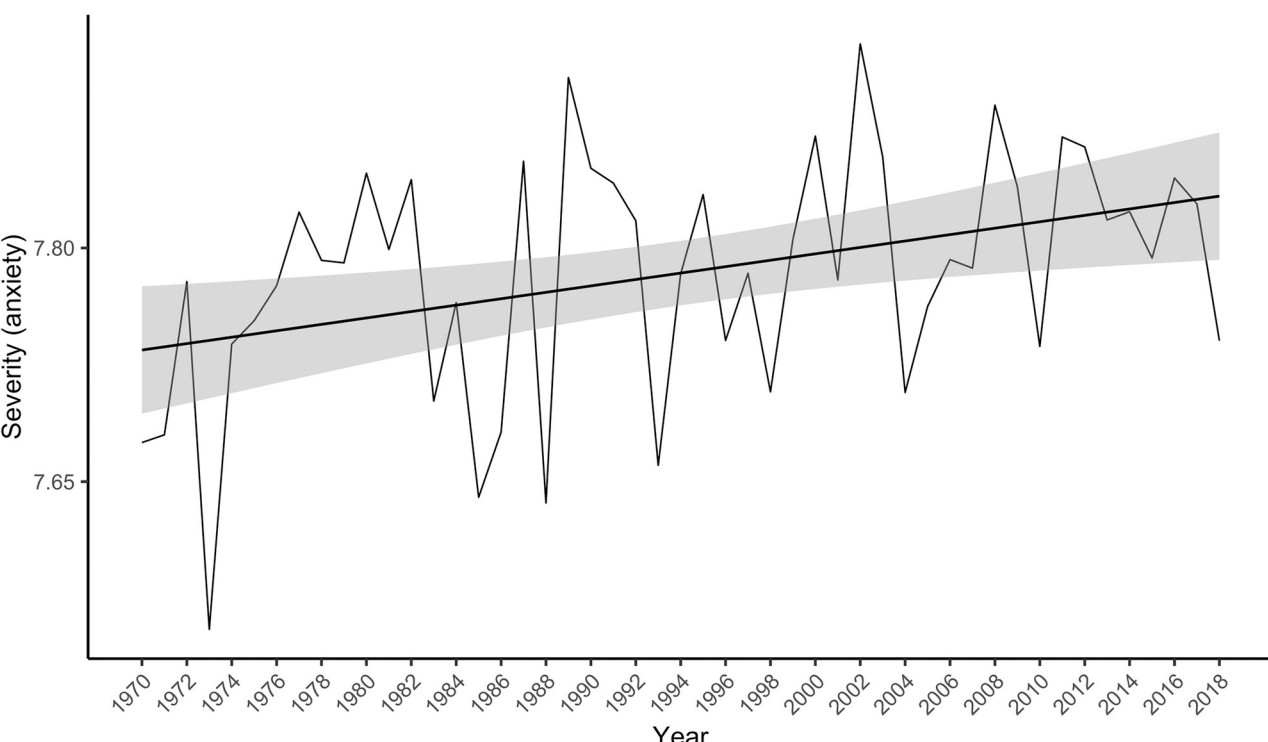

**Fig 6. The severity index for "anxiety" in the CoHA/CoCA corpus from 1970 to 2018.** The grey bars around the linear regression line indicate the standard error estimate.

collocates for either 'anxiety' or 'depression', 'disorder' became a popular collocate in the 2000s and 2010s. By implication, in both the academic and professional discourse of psychology and in the wider culture sampled by the general corpus, 'anxiety' and 'depression' were increasingly discussed in the context of pathology. That increase appears to have been gradual and lagged between professional and general discourse, and therefore cannot be confidently ascribed to a single historical event, such as the publication of DSM-III in 1980. Whether the rise in pathological language around 'anxiety' and 'depression' might be associated with

**Table 3. Top 10 anxiety collocates in the CoHA/CoCA corpus by decade.**

| 1970s | 1980s | 1990s | 2000s | 2010s |
|---|---|---|---|---|
| feel | feel | feel | depression | depression |
| man | fear | know | feel | people |
| know | time | fear | disorder | know |
| time | come | people | know | feel |
| fear | face | like | fear | like |
| like | great | think | people | think |
| ask | know | depression | time | fear |
| face | day | time | think | disorder |
| think | like | come | like | time |
| come | leave | attack | stress | come |

Words were ranked by their relative frequency in each decade, from highest (top row) to lowest.

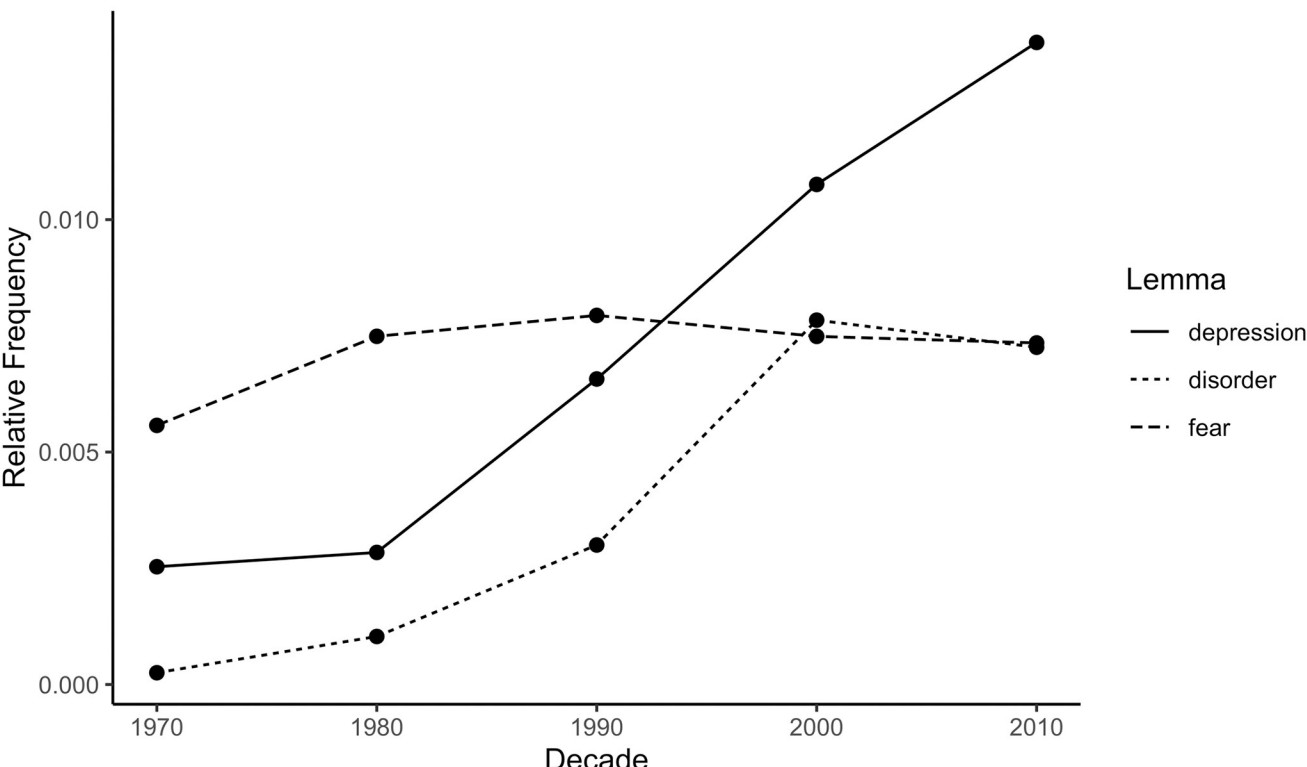

**Fig 7. Relative frequencies of selected anxiety collocates in the CoHA/CoCA corpus by decade.** Relative frequency is the summed repetitions of one lemma within a decade divided by the summed repetitions of all lemmas in the same decade. Larger relative frequency means higher frequency of the lemma in a particular decade.

historical increases in the prevalence of these conditions (e.g., [34]) remains to be determined. There may or may not be a relationship between the prevalence of a clinical phenomenon and the conditional probability of clinical terminology being used when it is mentioned.

Even more striking is the rising trend for the two concepts of interest to co-occur. In the psychology corpus, 'depression' became and remained the most common collocate of 'anxiety' in the 1980s, after not even entering the top 10 in the 1970s, and 'anxiety' became, and remained, the most common collocate for 'depression' in the same decade. This co-occurrence mirrors the substantial comorbidity and overlap of anxiety and depressive disorders [16, 18]. A similar convergence was evident in the general corpus, albeit appearing later. 'Depression' became the top collocate of 'anxiety' in the 2000s and 'anxiety' became its top collocate in the 2010s. The two concepts have clearly become a tightly bound pair in both the academic and general discourse. Because the two terms are high in emotional severity (i.e., affectively negative and intense), as are clinical terms like 'disorder' and 'symptom', at least part of the increase in the severity index over the study period may reflect the rising prominence of these collocations.

Although our findings do not support the predicted dilution of the meaning of 'anxiety' or 'depression', they are consistent with an increased pathologizing of these concepts over recent decades. Whereas 'anxiety' and 'depression' can refer to ordinary affective states, rather than to clinical conditions, and–judging from the collocates–largely did so in the 1970s and 1980s, the strong trend in both psychological and general discourse has been to place a clinical frame around them. That frame locates them in the context of diagnosis ('disorder' and 'symptom')

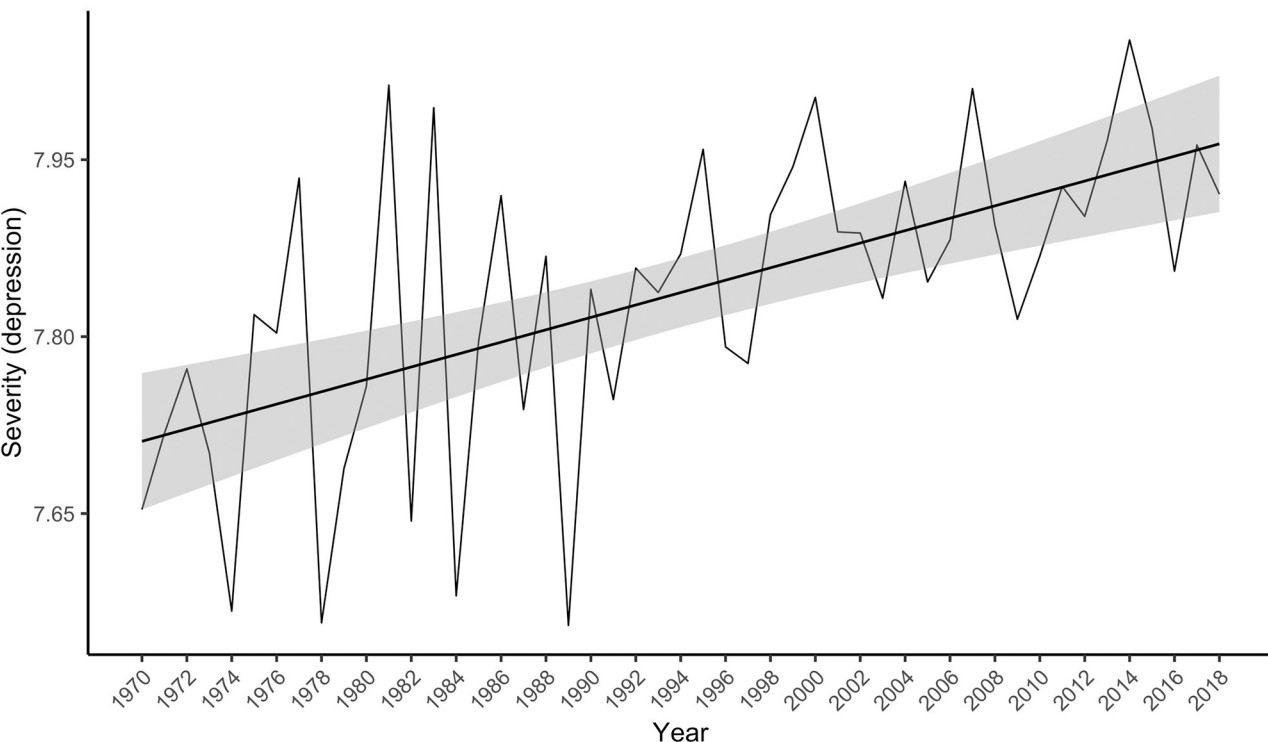

**Fig 8. The severity index for "depression" in the CoHA/CoCA corpus from 1970 to 2018.** The grey bars around the linear regression line indicate the standard error estimate.

rather than normal emotional distress and compounds them as linked pathological entities rather than as distinct experiences. This pathologizing trend appears to be the direct opposite of the normalizing trend (vertical concept creep) we predicted, but it may not be incompatible with it. It is possible that 'anxiety' and 'depression' are now being used to refer to less severe phenomena than in earlier times, but they are also used in a more clinical idiom than before. A tendency to use pathological or diagnostic language to make sense of everyday distress might

**Table 4. Top 10 depression collocates in the CoHA/CoCA corpus by decade.**

| 1970s | 1980s | 1990s | 2000s | 2010s |
|-------|-------|-------|-------|-------|
| time | year | people | people | anxiety |
| year | people | year | anxiety | know |
| war | day | know | know | people |
| day | time | time | year | like |
| know | like | suffer | disorder | year |
| think | woman | like | like | suffer |
| feel | think | think | suffer | disorder |
| find | help | anxiety | time | time |
| like | suffer | problem | think | life |
| life | find | war | problem | thing |

Words were ranked by their relative frequency in each decade, from highest (top row) to lowest.

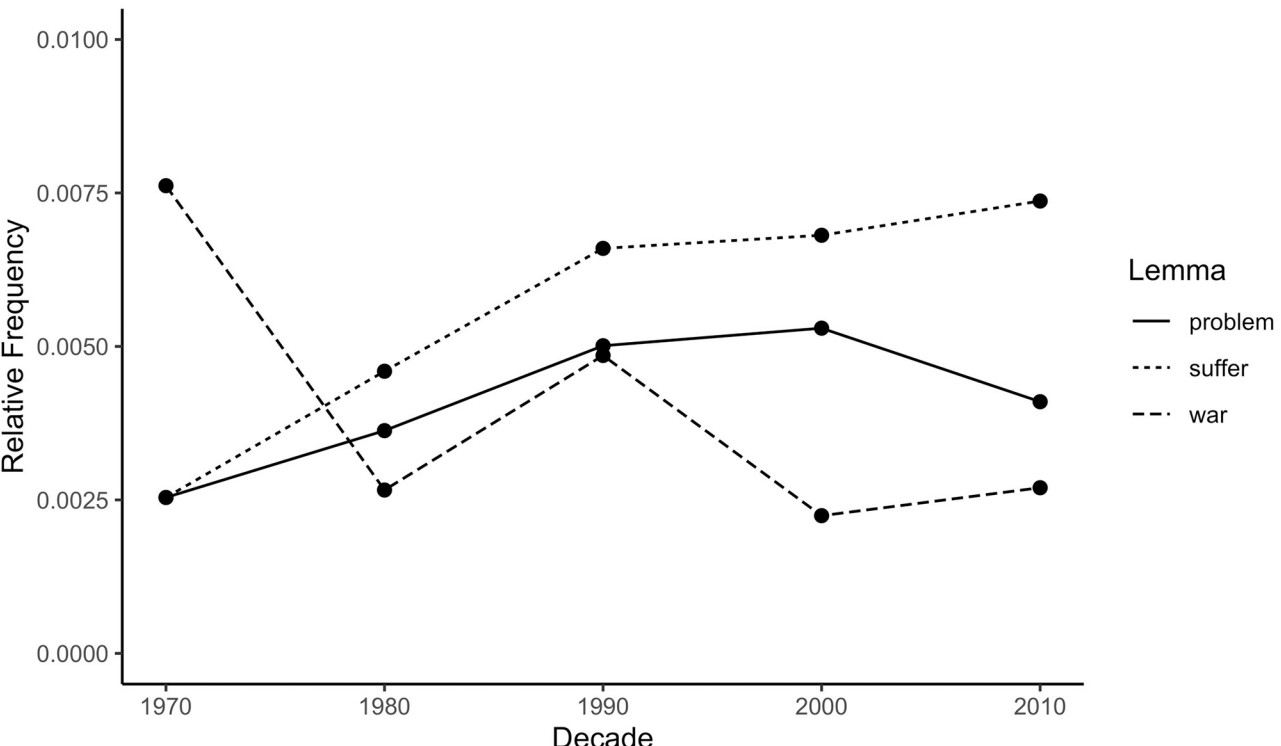

**Fig 9. Relative frequencies of selected depression collocates in the CoHA/CoCA corpus by decade.** Relative frequency is the summed repetitions of one lemma within a decade divided by the summed repetitions of all lemmas in the same decade. Larger relative frequency means higher frequency of the lemma in a particular decade.

be one way in which vertical creep takes place. Even so, the trends we have identified are better described as pathologizing rather than as vertical concept creep (normalizing).

The present research inevitably has limitations and weaknesses. Despite the breadth and size of the two text corpora, it is possible that unrelated historical changes in their composition might distort the severity trends we examined in our hypothesis tests (e.g., changes in the proportion of clinical psychology articles in the psychology corpus). It would also be inappropriate to conclude from the patterns observed in the general corpus how everyday people think about 'anxiety' and 'depression', seeing as the texts that compose the corpus are generated by an unrepresentative group of content producers (e.g., authors, journalists, bloggers). Our severity index, based on published norms of emotional meaning, is a readily automated way to assess the dimension of harm on which vertical concept creep takes place, from intensely negative to mild and innocuous, but it may not fully capture the complexity of harm, which may also have a moral component that is not reducible to affective intensity. It is also possible that some historical shifts have occurred in the connotations of words that might complicate the interpretation of the trends we observed, although we believe it is implausible that there has been a strong generalized tendency for word meanings to have increased in emotional severity over recent decades. Nevertheless, although the methodology has some potential limitations, the magnitude of our data sets and their great historical scope represent some compensating strengths. Future research might refine the methodology, examine additional text corpora including those drawn from social media, to explore media representations of mental illness, and determine whether similar patterns of pathologizing can be observed with other mental health-related concepts.

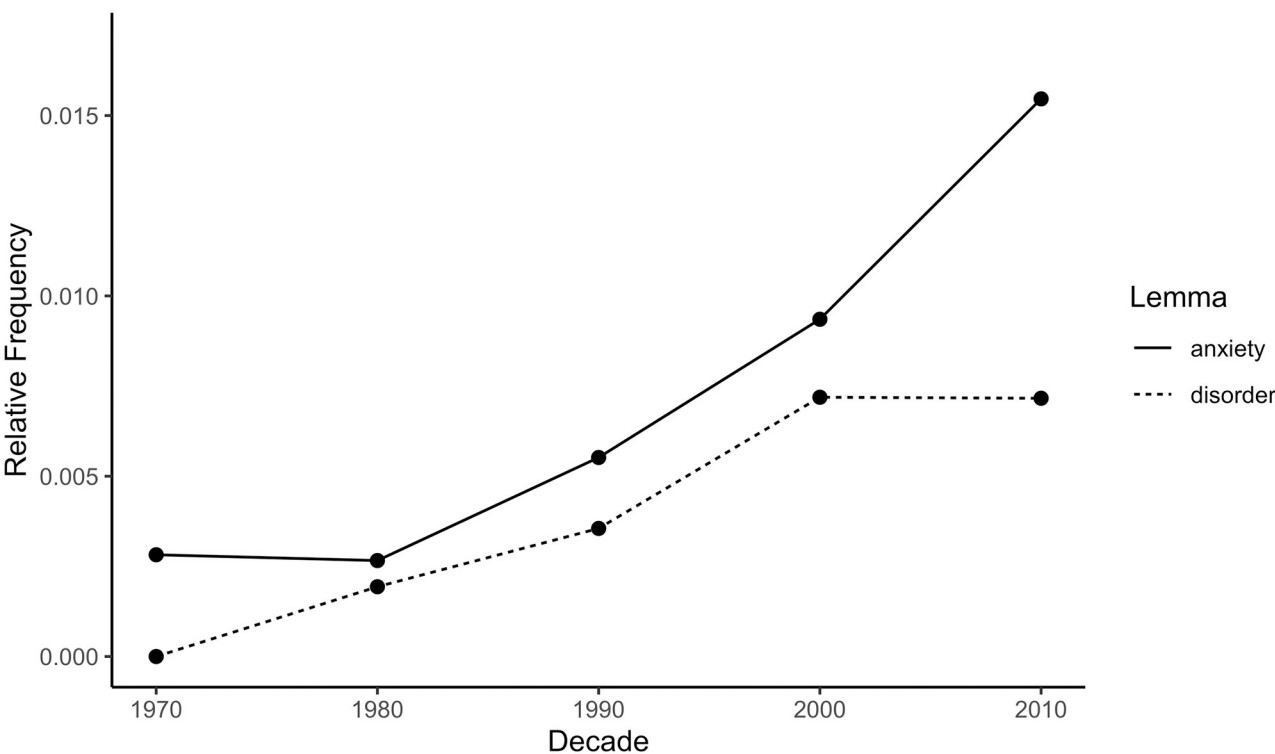

**Fig 10. Relative frequencies of selected depression collocates in the CoHA/CoCA corpus by decade.** Relative frequency is the summed repetitions of one lemma within a decade divided by the summed repetitions of all lemmas in the same decade. Larger relative frequency means higher frequency of the lemma in a particular decade.

## Conclusion

Using very large text corpora representing academic psychology and general culture (USA blogs, fiction, magazines, newspapers, spoken language, TV), we found that the concepts of 'anxiety' and 'depression' have undergone notable shifts in their emotional meaning. Contrary to the hypothesis that they would become increasingly associated with less intense, severe, or harm-related words, the opposite pattern consistently emerged across concepts and corpora. That pattern appeared to reflect, in part, a rising tendency to pathologize 'anxiety' and 'depression' by locating them in the semantic context of diagnosis, disorder, and symptoms. The cultural implications of this trend, and how it relates to the broad pattern of concept creep, remains to be determined.

## Acknowledgments

All corpus pre-processing and data extraction used Spartan, the University of Melbourne's general purpose hybrid high performance computing system: Lafayette, L., Sauter, G., Vu, L. and Meade, B., 2016. Spartan performance and flexibility: An hpc-cloud chimera. *OpenStack Summit*, *Barcelona*, *27*.

## Author Contributions

**Conceptualization:** Yu Xiao, Nick Haslam.

**Data curation:** Naomi Baes, Ekaterina Vylomova.

**Formal analysis:** Yu Xiao, Naomi Baes, Ekaterina Vylomova, Nick Haslam.

**Funding acquisition:** Nick Haslam.

**Investigation:** Yu Xiao, Naomi Baes, Ekaterina Vylomova.

**Methodology:** Yu Xiao, Naomi Baes, Ekaterina Vylomova, Nick Haslam.

**Project administration:** Yu Xiao, Naomi Baes, Ekaterina Vylomova, Nick Haslam.

**Resources:** Nick Haslam.

**Supervision:** Nick Haslam.

**Visualization:** Yu Xiao, Naomi Baes.

**Writing – original draft:** Yu Xiao, Nick Haslam.

**Writing – review & editing:** Yu Xiao, Naomi Baes, Ekaterina Vylomova, Nick Haslam.

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
