## [Decision Letter · Decision Letter 0]

16 May 2023

PONE-D-23-09235Have the concepts of ‘anxiety’ and ‘depression’ been normalized or pathologized? A corpus study of historical semantic changePLOS ONE

Dear Dr. Baes,

Thank you for submitting your manuscript to PLOS ONE. After careful consideration, we feel that it has merit but does not fully meet PLOS ONE’s publication criteria as it currently stands. Therefore, we invite you to submit a revised version of the manuscript that addresses the points raised during the review process.

We look forward to receiving your revised manuscript.

Kind regards,

Michal Ptaszynski, PhD

Academic Editor

PLOS ONE

Reviewers' comments:

Reviewer's Responses to Questions

**Comments to the Author**

1. Is the manuscript technically sound, and do the data support the conclusions?

Reviewer #1: Yes

Reviewer #2: Partly

2. Has the statistical analysis been performed appropriately and rigorously? 

Reviewer #1: Yes

Reviewer #2: No

3. Have the authors made all data underlying the findings in their manuscript fully available?

Reviewer #1: Yes

Reviewer #2: Yes

4. Is the manuscript presented in an intelligible fashion and written in standard English?

Reviewer #1: Yes

Reviewer #2: Yes

5. Review Comments to the Author

Reviewer #1: Thank you for the opportunity to review this interesting paper for PLoS One. The authors examined the shift in meaning of the words “anxiety” and “depression” by noting the frequency of words with which they were paired in two distinct but large corpora—one academic and one public. Hypothesizing that they would find a more casual use of the words developing from the 1970’s through the 2010’s, they were surprised to find the opposite: more collocation with words of a clinical nature, including with each other. There are just a few minor comments to consider before the paper is ready for publication in this Journal. As a disclaimer, language studies like this are not my area of expertise, but I give my comments as a generally interested clinical psychologist who does research on depression and anxiety.

Intro

Line 42-43: the bullying example is very illustrative and helpful for someone new to this type of study. Bravo.

Line 49-50: Implied is that vertical concept creep can only be “downward,” i.e. moving to include less severe phenomena as in the bullying example. Couldn’t vertical concept creep also include moving “upward” to include more severe phenomena, as the authors sort of found in the present study?

Lines 60-64: This argument is made with no citations. It would be stronger if there were some studies that at least hinted at some empirical evidence for the logic presented actually playing out.

Line 113-15: Why is it bad if people seek therapy, even if their “depression” is mild? In truth, it could be a public health strain on services etc., or people could end up on antidepressants with side effects that are unnecessary, but without explicating those implications the argument here sounds pretty weak.

Methods

Lines 181-83: I was confused as to why valence was summed with arousal. Wouldn’t it make more sense to multiply the two? That way a very positive emotion, like overwhelming joy, that causes a lot of arousal would be scored at the opposite end of the spectrum as, say, overwhelming fear. For ease of interpretability one could also make the valence numbers range from -4 to 4, rather than 1 to 9. Just a suggestion.

Results

No comments.

Discussion

One partial explanation for the rise in collocation of clinical words for anxiety and depression could be the rise in the actual phenomena of bona fide, DSM-5 depression and anxiety disorders. The authors did not discuss this possibility, but should. It is likely that the authors are correct that non-pathological, every day phenomena are referred to using clinical language more so today than in the 70’s or 80’s, but there are probably also more true clinical phenomena now than in the 70’s or 80’s as well. This should be considered.

Lines 326-9: A citation or two would be helpful to illustrate these “well established” concerns.

Line 348-51: While I agree with the gradual increase in anxiety’s and depression’s collocation with clinical terms when viewed decade by decade, the increase in severity index scores of the collocated words (figure 1 especially, for anxiety) suggests that the only real shift occurred from 1985-1993 or so. Do the authors have any explanation for this?

Line 370: Should probably read, “This pathologizing trend appears to be…” just for style purposes.

I sign this review, as is my practice, that my identity will be included with it as long that is acceptable to the journal.

Daniel Norton

Gordon College

Reviewer #2: Review of PONE-D-23-09235

This manuscript describes an analysis of collocates of the terms "anxiety" and "depression" in historical corpora to test the hypothesis that the meaning of these concepts have broadened over time. This is an interesting study, but I have several questions and concerns for the authors to clarify.

1. The terms "emotional intensity" and "semantic severity" seem to be used interchangeably, but I am not sure if they truly reflect the same concept.

2. An key assumption that the authors' methodology relies on is that the valence and arousal of words (as measured by Warriner et al.) do not change over time. Is it reasonable to assume that the valence and arousal value of words are static and constant over such a long time period given that many studies find that word meanings do evolve and change over history? E.g., the concept "gay" has a more positive connotation earlier in history, but one could imagine that it currently has a more negative connotation in modern times.

3. Why was horizontal concept creep not explored in this paper? This was surprisingly after such a detailed description of the two types of concept creep.

4. Many important details about the methodology, materials are not provided. For example:

- In the psychology corpus, how are the journals distributed in terms of subfield of psychology? Are these mostly clinical psychology journals? What about psychiatry journals which would have a large coverage of research in depression in anxiety?

- How evenly distributed are the corpora in terms of the date range of 1970 to 2018? Do different years provide roughly the same number of words or is it skewed in some years? How frequently do the terms "anxiety" and "depression" occur in the corpus and how is this similar or different across the time period assessed?

- There is no information (as far as I can tell) about how many of the collocates *do not* have any valence or arousal norms in the Warriner dataset. What is the level of missingness and would this have any effect on the results?

5. The severity index is an interesting measure but it is simply a summation of the valence and arousal scores of the word and some information loss may occur. For instance, it is possible for two words to have the same summed index score, but for one word this value is driven by negative valence, and for the other word this value is mostly driven by high arousal. How can such an index distinguish the relative contributions of valence and arousal to the computation of "severity"?

6. Some questions about the analysis approach for the authors to consider:

- First of all, it would be better if the figures were provided in-text and not on the OSF only. Based on those figures, it seemed like a linear model may not be appropriate given the clearly non-linear trends in the data. Perhaps a Generalized Additive Model (GAM) is more appropriate for capturing non-linearities in the data.

- It is not clear how the predictor "year" is represented in the model? Was it mean-centered, contrast coded in some way or just included as a continuous variable (which is not recommended)? The authors may want to look into techniques that can analyze time-series data.

7. Finally, the fact that "anxiety" and "depression" are themselves highly frequent collocates of each other really makes it quite challenging to understand and interpret the results. I find it surprising that these concepts are not "high in emotional severity" (p. 20)... What is the severity for these words? Perhaps a more detailed frequency analysis would be useful to help us understand what the results mean - for instance, could the collocates be analyzed for their /relative change/ in frequency over time to explore what concepts are main drivers of the unexpected rise in emotional severity?

6. PLOS authors have the option to publish the peer review history of their article (what does this mean?). If published, this will include your full peer review and any attached files.

Reviewer #1: **Yes: **Daniel J Norton, Ph.D.

Reviewer #2: No

---

## [Author Response · Author response to Decision Letter 0]

1 Jun 2023

Response to Reviewers

We thank the reviewers for their very helpful feedback. We have seriously engaged with all of their concerns and believe that the manuscript has been significantly improved as a result. Please find our responses below in response to each critical comment. We quote each comment in full, excluding remarks that did not request clarification or expressed agreement or positive evaluation of the manuscript.

Reviewer 1

1. “Line 49-50: Implied is that vertical concept creep can only be “downward,” i.e. moving to include less severe phenomena as in the bullying example. Couldn’t vertical concept creep also include moving “upward” to include more severe phenomena, as the authors sort of found in the present study?”

We now clarify this point with an added sentence which agrees that concepts can indeed shift in the other direction, but notes that such a shift does not constitute concept creep in our sense because by definition concept creep involves an expansion rather than a contraction of concept meaning. Generally, also, the harm-related concepts studied in research on concept creep cannot easily expand vertically because there is an existing severity maximum that does not change (e.g., for ‘trauma’ it is the most traumatic experience possible). The threshold for deciding when the concept applies might move upward (i.e., a more stringent definition of trauma’) but that narrows the meaning of trauma because the maximum does not also rise.

2. “Lines 60-64: This argument is made with no citations. It would be stronger if there were some studies that at least hinted at some empirical evidence for the logic presented actually playing out”.

We have now added three citations to back up our argument in lines 60-64. Dakin et al. have empirically documented the trivialization effect, Jones & McNally have shown that broadened trauma concepts can increase negative responses to disturbing events, and Furedi presents qualitative evidence for the rise of vulnerability associated with rising sensitivity to harm.

3. “Line 113-15: Why is it bad if people seek therapy, even if their “depression” is mild? In truth, it could be a public health strain on services etc., or people could end up on antidepressants with side effects that are unnecessary, but without explicating those implications the argument here sounds pretty weak.”

We did not mean to suggest that it was inappropriate for people with mild depression to seek treatment, and regret that implication. Our point is simply that if many people were to self-overdiagnose (e.g., seek or demand treatment when they are not clinical depressed), this might in some cases lead to adverse consequences. We have added a sentence that includes two of the consequences the reviewer suggests, along with another that is backed by a new citation (see lines 118-121).

4. “Lines 181-83: I was confused as to why valence was summed with arousal. Wouldn’t it make more sense to multiply the two? That way a very positive emotion, like overwhelming joy, that causes a lot of arousal would be scored at the opposite end of the spectrum as, say, overwhelming fear. For ease of interpretability one could also make the valence numbers range from -4 to 4, rather than 1 to 9. Just a suggestion”

We believe summing is the appropriate way of combining the two indices, and it has been carried out in a previous published study using the method. Multiplying the valence and arousal indices would create a very skewed severity measure if we do not re-scale the indices (range 1 to 81 where neutrality is 25) and multiplying the re-scaled (-4 to +4) indices would be inappropriate because a low intensity positive word would get the same score as a high intensity negative one. We believe summing is a simple method that captures emotional severity well by scoring low intensity positive words (e.g., “contented”) lowest, high intensity negative words highest, and low arousal negative and high arousal positive words intermediate. Regarding possibly re-scaling the valence and arousal indices, we would prefer to retain the existing 109 scale, which is based directly on the published norms themselves.

5. “One partial explanation for the rise in collocation of clinical words for anxiety and depression could be the rise in the actual phenomena of bona fide, DSM-5 depression and anxiety disorders. The authors did not discuss this possibility, but should. It is likely that the authors are correct that non-pathological, every day phenomena are referred to using clinical language more so today than in the 70’s or 80’s, but there are probably also more true clinical phenomena now than in the 70’s or 80’s as well. This should be considered.”

We agree that there may have been some historical increase in the prevalence of anxiety and depressive disorders over the decades of our study. However, we believe any such rise in prevalence would more likely manifest in a rise in the frequency with which “anxiety” and “depression” appear in our text corpora rather than in the <relative> frequency of clinically-related versus other collocates when these words do appear. The fact that there is more clinical anxiety present now than in 1970 should not obviously change the kinds of language that is used around the word “anxiety” now versus then. Clinical anxiety (and probably subclinical anxiety too) may have been less prevalent and talked about then, but it is not clear to us that <when> it was talked about clinical terminology should be less common just because the objective prevalence was lower. Nevertheless, we agree that this is a possibility and have added two sentences to this effect at the end of the third paragraph of the Discussion. Interestingly, our new Figure 1 (see below) shows a substantial rise in the frequency of “anxiety” and “depression” in the psychology corpus.

6. “Lines 326-9: A citation or two would be helpful to illustrate these “well established” concerns.”

We have added three relevant citations here, as requested.

7. “Line 348-51: While I agree with the gradual increase in anxiety’s and depression’s collocation with clinical terms when viewed decade by decade, the increase in severity index scores of the collocated words (figure 1 especially, for anxiety) suggests that the only real shift occurred from 1985-1993 or so. Do the authors have any explanation for this?”

We do not have a post hoc explanation for the possible difference in the timing of these trends, but note that changes in the overall severity index are driven by a large and diverse assortment of collocates and the specific clinical terms mentioned in the paragraph are likely to be only one modest component of any such changes. We do not claim in the manuscript that shifts in the relative prominence of “disorder” and “symptom” as collocates of “anxiety” and “depression” are primarily responsible for the rising trajectory of the severity indices.

8. “Line 370: Should probably read, “This pathologizing trend appears to be…” just for style purposes.”

We have made this change as requested.

Reviewer 2

1. “The terms "emotional intensity" and "semantic severity" seem to be used interchangeably, but I am not sure if they truly reflect the same concept.”

We were intending to use “emotional intensity” (used 3 times), and “emotional severity” (7 times) and “semantic severity” (9 times) to express the same concept, which our index aims to capture. In essence, we are assessing the extent to which words have meanings that are emotionally negative and high in arousal. “Emotional intensity” is not an entirely satisfactory term because it could in principle refer to positive emotion as well as negative. “Semantic severity” is not perfect because it doesn’t specify the dimension(s) on which severity is being assessed, although severity is a key aspect of the concept. In response to the reviewer’s concern, we therefore now consistently use “emotional severity” throughout the manuscript and refer to the index itself simply as the “severity index”.

2. “A key assumption that the authors' methodology relies on is that the valence and arousal of words (as measured by Warriner et al.) do not change over time. Is it reasonable to assume that the valence and arousal value of words are static and constant over such a long time period given that many studies find that word meanings do evolve and change over history? E.g., the concept "gay" has a more positive connotation earlier in history, but one could imagine that it currently has a more negative connotation in modern times.”

We do not believe that our methodology relies on the valence and arousal of words being unchanging. No doubt individual words are subject to some shifts in valence- and arousal-related connotations over periods of decades, although we suspect major changes are likely to be rare. We note that all of our severity trend findings are based on (weighted) average mean severity scores of thousands of unique words, so these trends could only be invalid if changes in the connotations of words are consistently and strongly occurring in one direction. That is, our findings could only occur due to changes in word connotations if words in general are becoming markedly more negative and high arousal over time, a possibility we find implausible. Nevertheless, we have added the following sentence to the penultimate paragraph of the Discussion to acknowledge this issue: “It is also possible that some historical shifts have occurred in the connotations of words that might complicate the interpretation of the trends we observed, although we believe it is implausible that there has been a strong generalized tendency for word meanings to have increased in emotional severity over recent decades.”

3. “Why was horizontal concept creep not explored in this paper? This was surprisingly after such a detailed description of the two types of concept creep.”

Horizontal creep was not explored in the current manuscript for two reasons. First, there have been numerous studies of horizontal creep of a range of harm-related concepts using computational linguistic methods (see below for references), but only one previous study of vertical creep of a single concept using the current method. Second, the issue of pathologization that we focus on in the manuscript is primarily to do with vertical rather than horizontal creep, specifically the encroachment of pathology-related language on “normal” (less severe) phenomena. We believe a focused investigation on vertical creep is appropriate for these reasons.

Haslam, N., Vylomova, E., Zyphur, M., & Kashima, Y. (2021). The cultural dynamics of concept creep. American Psychologist, 76(6), 1013–1026.

Vylomova, E., & Haslam, N. (2021). Semantic changes in harm-related concepts in English. In N. Tahmasebi, L. Borin, A. Jatowt, Y. Xu & S. Hengchen (Eds.), Computational approaches to semantic change (pp. 93-121). Language Science Press.

Vylomova, E., Murphy, S., & Haslam, N. (2019). Evaluation of semantic change of harm-related concepts in psychology. In Proceedings of the 1st International Workshop on Computational Approaches to Historical Language Change, 29–34.

4. “Many important details about the methodology, materials are not provided. For example:

- In the psychology corpus, how are the journals distributed in terms of subfield of psychology? Are these mostly clinical psychology journals? What about psychiatry journals which would have a large coverage of research in depression in anxiety?

- How evenly distributed are the corpora in terms of the date range of 1970 to 2018? Do different years provide roughly the same number of words or is it skewed in some years? How frequently do the terms "anxiety" and "depression" occur in the corpus and how is this similar or different across the time period assessed?

- There is no information (as far as I can tell) about how many of the collocates *do not* have any valence or arousal norms in the Warriner dataset. What is the level of missingness and would this have any effect on the results?”

a) The 875 journals in the psychology corpus are not mostly clinical psychology journals. These make up 171 (19.5%) of the journal set, which is distributed across all subfields of psychology. The 875 journals were all those listed under “psychology” in the E-Research and PubMed databases and therefore are broad in scope. We have added a short description of the main groupings of journals according to their Scimago classification in the Materials section of the Method. Psychiatry journals were not picked up using this search process unless also tagged as “psychology” in the relevant databases. 

b) The psychology corpus contains substantially fewer abstracts in the early years than in later ones, reflecting the explosion of psychology publication over the last half century. As noted in the Method, we began the study period in 1970 because prior to then the data were sparse. The general corpus has a much more even distribution of words. We emphasize that the size of the corpus in a particular year should have no systematic relationship with the mean severity score for that year, merely increasing the error around the mean (e.g., observe the greater jaggedness of the plots in the earlier years for the psychology corpus). We have now added a graph (Figure 1) presenting the relative frequency of “anxiety” and “depression” in the two corpora.

c) The Warriner norms cover almost 14,000 common English lemmas. Any material in the corpora that did not match to one of these normed lemmas is likely to reflect very infrequent words. Although it is impossible to determine whether these “missing” (i.e., unmatched) lemmas would alter the severity trends because there is no way to evaluate their severity, we believe it is implausible because a large majority of the collocates are non-missing lemmas, representing the most common words in English and in the corpora. We have now added to the results section the proportion of each set of collocates that matched to the Warriner norms (i.e., that were non-missing), and these show that more than 70% of the collocates matched the norms. We thank the reviewer for raising this issue and believe these new data add confidence to our findings.

5. “The severity index is an interesting measure but it is simply a summation of the valence and arousal scores of the word and some information loss may occur. For instance, it is possible for two words to have the same summed index score, but for one word this value is driven by negative valence, and for the other word this value is mostly driven by high arousal. How can such an index distinguish the relative contributions of valence and arousal to the computation of "severity"?

The index cannot distinguish the relative contribution of its two components, but that contribution should tend to be approximately equal given that two variables measured on the same scale with roughly equal variability are being summed. Our goal in creating the summed index was to capture the concept of emotional severity, understood as involving connotations of high arousal and negative valence, not to examine each component separately. We aimed to assess collocates on a continuum from “emotionally positive and calm” to “unpleasant and intense”, and the fact that some intermediate scoring collocates might be positive and high arousal (e.g., excited), negative and low arousal (bored), or neutral and average arousal is therefore not problematic. This issue of possible heterogeneity among midrange cases arises for all indices that sum imperfectly correlated variables. We therefore believe it is legitimate to use our summed index for its intended purpose.

6. “Some questions about the analysis approach for the authors to consider: - First of all, it would be better if the figures were provided in-text and not on the OSF only. Based on those figures, it seemed like a linear model may not be appropriate given the clearly non-linear trends in the data. Perhaps a Generalized Additive Model (GAM) is more appropriate for capturing non-linearities in the data.

- It is not clear how the predictor "year" is represented in the model? Was it mean-centered, contrast coded in some way or just included as a continuous variable (which is not recommended)? The authors may want to look into techniques that can analyze time-series data.”

a) It was our intention for the figures to be displayed in the manuscript. Please accept our apologies, as it seems the editorial system did not display the TIFF files well in the pdf version. We have attempted to rectify this. Having said that, we believe the trend graphs are generally reasonably close to linear (except Figure 2). As the hypotheses we tested in the analyses were simple ones – historical declines in severity index – and our goal was not to model the trends in greater detail, we believe it is appropriate to run the simplest analyses. Despite some possible nonlinearities, we believe the rising trends are unmistakeable in every figure.

b) For the same reason, we used year as a continuous variable in the simple regression (essentially a correlation) and believe this is appropriate for the simple analytic purpose of testing for a historical rise or fall. We agree that time series analysis would be appropriate if we were carrying out more complex analyses involving predictors of the trend, exploration of endogenous factors/autocorrelation underlying the trend, forecasting the trend, and so on.

7. “Finally, the fact that "anxiety" and "depression" are themselves highly frequent collocates of each other really makes it quite challenging to understand and interpret the results. I find it surprising that these concepts are not "high in emotional severity" (p. 20)... What is the severity for these words? Perhaps a more detailed frequency analysis would be useful to help us understand what the results mean - for instance, could the collocates be analyzed for their /relative change/ in frequency over time to explore what concepts are main drivers of the unexpected rise in emotional severity?”

In fact, “anxiety” and “depression” are both high in emotional severity, rated 11.4 and 11.8 respectively compared to a mean for all collocates of 7.9 (these figures are presented in the Results). The sentence in question was badly punctuated: instead of “Because the two terms are not surprisingly high in emotional severity” it should have read “Because the two terms are, not surprisingly, high in emotional severity”. We apologize for this error and have corrected it. In view of this, we believe our figures showing historical rises in the frequency of several high severity collocates helps to explain the unexpected rises we observed.

Again, we thank both reviewers for their very careful attention to our manuscript and believe that our responses have made the work stronger.

---

## [Decision Letter · Decision Letter 1]

19 Jun 2023

Have the concepts of ‘anxiety’ and ‘depression’ been normalized or pathologized? A corpus study of historical semantic change

PONE-D-23-09235R1

Dear Dr. Baes,

We’re pleased to inform you that your manuscript has been judged scientifically suitable for publication and will be formally accepted for publication once it meets all outstanding technical requirements.

Kind regards,

Michal Ptaszynski, PhD

Academic Editor

PLOS ONE

Additional Editor Comments (optional):

Reviewers' comments:

Reviewer's Responses to Questions

**Comments to the Author**

1. If the authors have adequately addressed your comments raised in a previous round of review and you feel that this manuscript is now acceptable for publication, you may indicate that here to bypass the “Comments to the Author” section, enter your conflict of interest statement in the “Confidential to Editor” section, and submit your "Accept" recommendation.

Reviewer #1: All comments have been addressed

Reviewer #2: All comments have been addressed

2. Is the manuscript technically sound, and do the data support the conclusions?

Reviewer #1: Yes

Reviewer #2: (No Response)

3. Has the statistical analysis been performed appropriately and rigorously? 

Reviewer #1: Yes

Reviewer #2: (No Response)

4. Have the authors made all data underlying the findings in their manuscript fully available?

Reviewer #1: Yes

Reviewer #2: (No Response)

5. Is the manuscript presented in an intelligible fashion and written in standard English?

Reviewer #1: Yes

Reviewer #2: (No Response)

6. Review Comments to the Author

Reviewer #1: The authors did a good job addressing all of my concerns. I recommend the paper be accepted, and appreciate the opportunity to review this interesting paper.

Reviewer #2: (No Response)

7. PLOS authors have the option to publish the peer review history of their article (what does this mean?). If published, this will include your full peer review and any attached files.

Reviewer #1: No

Reviewer #2: No

---

## [Editor Report · Acceptance letter]

21 Jun 2023

PONE-D-23-09235R1 

Have the concepts of ‘anxiety’ and ‘depression’ been normalized or pathologized? A corpus study of historical semantic change 

Dear Dr. Baes:

I'm pleased to inform you that your manuscript has been deemed suitable for publication in PLOS ONE. Congratulations! Your manuscript is now with our production department. 

Kind regards, 

on behalf of

Dr. Michal Ptaszynski 

Academic Editor

PLOS ONE